# The impact of primary colonizers on the community composition of river biofilm

**Roshan Angoshtari[1], Kim T. Scribner[2], Terence L. Marsh[1]***

**1** Department of Microbiology and Molecular Genetics, Michigan State University, East Lansing, MI, United States of America, **2** Department of Fisheries and Wildlife, Michigan State University, East Lansing, MI, United States of America

* marsht@msu.edu

**Data Availability Statement:** Sequence reads have been deposited in the NCBI Sequence Read Archive (BioProject ID PRJNA849629).

**Funding:** Funding sources included the Michigan Department of Natural Resources State Wildlife

## Abstract

As a strategy for minimizing microbial infections in fish hatcheries, we have investigated how putatively probiotic bacterial populations influence biofilm formation. All surfaces that are exposed to the aquatic milieu develop a microbial community through the selective assembly of microbial populations into a surface-adhering biofilm. In the investigations reported herein, we describe laboratory experiments designed to determine how initial colonization of a surface by nonpathogenic isolates from sturgeon eggs influence the subsequent assembly of populations from a pelagic river community, into the existing biofilm. All eight of the tested strains altered the assembly of river biofilm in a strain-specific manner. Previously formed isolate biofilm was challenged with natural river populations and after 24 hours, two strains and two-isolate combinations proved highly resistant to invasion, comprising at least 80% of the biofilm community, four isolates were intermediate in resistance, accounting for at least 45% of the biofilm community and two isolates were reduced to 4% of the biofilm community. Founding biofilms of *Serratia* sp, and combinations of *Brevundimonas* sp.-*Hydrogenophaga* sp. and *Brevundimonas* sp.-*Acidovorax* sp. specifically blocked populations of *Aeromonas* and *Flavobacterium*, potential fish pathogens, from colonizing the biofilm. In addition, all isolate biofilms were effective at blocking invading populations of *Arcobacter*. Several strains, notably *Deinococcus* sp., recruited specific low-abundance river populations into the top 25 most abundant populations within biofilm. The experiments suggest that relatively simple measures can be used to control the assembly of biofilm on the eggs surface and perhaps offer protection from pathogens. In addition, the methodology provides a relatively rapid way to detect potentially strong ecological interactions between bacterial populations in the formation of biofilms.

## Introduction

Both freshwater and marine systems are under siege on multiple fronts by the activities of *Homo sapiens*. Many of these activities can alter fundamental properties of these ecosystems, including the microbial community [1–5]. We describe the microbial community as a "fundamental property" because of the ubiquity and essential functions that these communities

Grants Program T-10-T-5 Study 237026 to KS (https://www.michigan.gov/dnr/buy-and-apply/grants/aq-wl/wildlife-hab), Center for Water Sciences Water Cube initiative at MSU to TLM & KS (https://water.msu.edu/watercube/), and the Department of Microbiology and Molecular Genetics at MSU (TLM). In addition, we thank the Department of Microbiology and the College of Natural Science for partial funding including a Thesis Completion grant (RA). The funders had no role in study design. data collection and analysis, decision to publish, or preparation of the manuscript.

**Competing interests:** The authors have declared that no competing interests exist.

perform. Aquatic metazoans have adapted to and evolved with aquatic microbes over millions of years [6–10]. Sudden changes to the structure of these microbial communities can cause dramatic shifts in successful recruitment and survival of allied species, eg. amphibia, fish and invertebrates. A critical example of the impact that compositional shifts in the aquatic microbial communities can have is on the survival of eggs of aquatic invertebrates and vertebrates. In many cases, these eggs are expelled into the aquatic milieu as aseptic structures that are rapidly colonized by populations of the microbial community, forming a biofilm on the egg's surface. Under these circumstances, the phylogenetic composition of the biofilm can dictate survival rates of the eggs and a community that has been radically shifted in response to environmental change, can reduce this rate. Perhaps the most pervasive example of this is eutrophication which has become a global problem [11–19]. The hallmark of eutrophication is a shift in the microbial community to greater numbers of phytoplankton in response to an increase in nutrients, primarily nitrogen and phosphorus, into the water. The consequences of this simple shift can be profound as the upper regions of a lake or river can become hyper oxygenated during the day and anoxic during the night, with larger diurnal shifts between these two conditions than seen in oligotrophic systems [20, 21]. Moreover, an increase in biomass can lead to an excess in decaying organic matter in the benthos that can become increasingly anoxic and less hospitable to the fertilized eggs of broadcast spawners like lake sturgeon. Shifts in environmental conditions can also favor *Saprolegnia* spp., which have become a widespread and economically costly predator of fish eggs in hatcheries [22–24]. There are of course other predators of eggs, but the primary community for which defenses must be established, is microbial.

Our investigations into the biofilm that assembles on the surface of Sturgeon eggs began over a decade ago and was stimulated by the high levels of egg mortality detected in Lake Sturgeon (*Ascipenser fluvescens)*, a considerable portion of which were thought attributable to microbial activity. These initial studies revealed that i.) the egg-associated bacterial community differed substantially from the bacterial community of the river [25, 26], ii.) the egg-associated microbiomes on healthy and moribund eggs were different [26, 27], iii.) the number of bacteria on the egg was proportional to time and the source of water [28], iv.) the bacterial community on the egg changes over time [26, 27], and v.) egg mortality is directly related to the phylogenetic structure of the aquatic bacterial community [26, 27]. Two examples of this last point are the attack of amphibian eggs in the wild by *Batrachochytrium dendrobatidis* [29, 30] and the global problem of saprolegniasis in the hatchery. The spread of *B. dendrobatidis* is thought to be a combination of factors including climate shifts and the international trade in amphibians [31, 32] while saprolegniasis can be pervasive in hatcheries where crowding and degraded environmental conditions are prevalent [33–36]. There is evidence that in infections with either *B. dendrobatidis* or *Saprolegnia* spp., the native microbiome can play a constructive role in defense [37–40]. Preliminary studies also revealed that intervention early in the assembly of the egg-associated microbial community could have positive effects on egg mortality [25]. These investigations led to a closer examination of interactions between bacterial isolates from the egg when forming biofilms [41], so that these specific interactions might be used to minimize infections in the hatchery by manipulating the assembly of the egg microbiome.

The investigations reported herein focused on isolates from the sturgeon egg [41] and asked the simple question of whether established biofilm of these isolates could influence subsequent colonization by wild bacterial populations. Several of the isolates were identified as putative probiotics based on their presence and prevalence on healthy eggs (in contrast to the communities of moribund eggs) as well as the identification of antimicrobial activities associated with several of the strains [41]. We hypothesized that each established biofilm comprised of an individual isolate would influence developing river biofilm uniquely. Moreover, previous

work had shown that mixed biofilms established with three of our isolates, *Brevundimonas* in combination with either *Hydrogenophaga* or *Acidovorax*, had properties that surpassed the biofilms of individual isolates [42]. The suggestion of synergistic interactions within these two-isolate biofilms prompted their inclusion in the study. To test these hypotheses single and double isolate biofilms were established in microtiter plates and then challenged with a river bacterial community comprised of hundreds of populations. The results indicated that all isolates were capable of blocking select river populations from invading established biofilms and that some isolate biofilms specifically blocked putative fish pathogens. The data are discussed with respect to the hatchery management goal of reduced mortality by establishing a beneficial, rather than an antagonistic biofilm on the egg's surface, and as an approach for investigating strain-strain interactions.

## Materials and methods

### Strains, media and cultivation

*Pseudomonas aeruginosa* PAO1 was the generous gift of Dr. Michael Bagdasarian at Michigan State University. This strain was used as a strong positive control for biofilm formation and other strains of this species are routinely isolated from freshwater sources [43–46] including the Red Cedar River (A. Chan & T.L. Marsh, unpublished), the source of river water for these studies. Additional isolates in this study were purified from the eggs of lake sturgeon *Acipenser fulvescens* [41], or in the case of *Serratia marcescens* RL-10, from soil [47] *Serratia marcescens* RL-10 was included because of our experience with this strain in forming biofilm [47] and its ability to produce antimicrobial compounds. Isolates from the sturgeon egg were initially identified with partial (300–400 bases) 16S rRNA sequencing [41] and later with nearly complete 16S rRNA sequence [27]. The genus affiliations and accession numbers are *Acidovorax* F19: MH465527, *Bacillus* C20: KY075739, *Brevundimonas* F16: MH465526, *Deinococcus* F4: KY075762, *Hydrogenophaga* F14: MH465525, *Serratia* D14:KY075703 and *Serratia marcescens* RL-10:MF581051. Genomic sequence analysis [27] identified *Acidovorax* F19 as closest to *Acidovorax* sp. CF316 (Genbank accession # AKJX00000000) and *Hydrogenophaga* F14 as closest to *Hydrogenophaga* sp. RAC07 (Genbank accession # CP016449). Initially, *Brevundimonas* F16 was identified as a *Caulobacter* [41], but more complete 16S rRNA sequence and a genome sequence [27] have identified this isolate as a *Brevundimonas*, closest to *Brevundimonas subvibrioides* strain 32-68-21 (Genbank accession # NCEQ00000000). The genomic comparisons were made with Mash/MinHash as implemented in Patric 3.6.7 [48]. The isolates were representative of the culturable phylogenetic diversity found on the egg. In nature, the bacterial communities on eggs are diverse with hundreds of detectable operational taxonomic units. In addition, these isolates were of particular interest because of previous characterizations of biofilm formation and resilience [41, 42]. All isolates were known to produce biofilm at levels that were strain-dependent under our test conditions [41]. Preliminary work with *Acidovorax* colonization of freshly fertilized sturgeon eggs suggested that egg survival in a hatchery setting was improved (26). *Bacillus* C20 and *Serratia* D14 inhibited growth of *Flavobacterium* sp. C05, *F. columnare* 090702–1 and *Aeromonas* salmonicida 05100658A59, known fish pathogens, in agar overlay tests [41]. Furthermore, we knew that established biofilms of *Acidovorax* F19, *Brevundimonas* F16 and *Hydrogenophaga* F14 were disrupted by a *Pseudomonas* strain commonly found in freshwater, including the Red Cedar River [41]. *Serratia* D14 was particularly aggressive in soft agar overlay tests against four of the 25 tested isolates from sturgeon eggs, including *Hydrogenophaga* F14, which was among the most sensitive in these tests [41]. In addition, combinations of *Brevundimonas* and *Acidovorax* or *Brevundimonas* and *Hydrogenophaga* produced more biofilm when paired than single species biofilms [42].

Finally, single species biofilms of these three strains had reduced metabolic activity when challenged with tobramycin but the mixed species biofilm of *Acidovorax* and *Brevundimonas* had increased metabolic activity when tobramycin was present [42]. These attributes of the seven isolates from sturgeon eggs led to their selection for this study.

Bacterial strains were maintained at -80˚C on 20% (v/v) glycerol. All strains were revived at 25˚C on R2A agar (Difco) for two days before each experiment and subsequently a single colony was inoculated into R2Broth (Yeast extract 0.5g/L, Proteose Peptone No. 3 0.5g/L, Casamino Acids 0.5g/L, Dextrose 0.5g/L, Soluble Starch 0.5g/L, Sodium Pyruvate 0.3g/L, Dipotassium Phosphate 0.3g/L, Magnesium Sulfate 0.05g/L). Overnight cultures (16 hours) grown at 25˚C in a tube on a rotating rack were used as inoculants into microtiter plates.

River water obtained from the Red Cedar River in East Lansing MI served as the source of natural environmental bacterial populations. This watershed is designated as Hydrologic Unit Code 04050004 by the United States Geological Survey. The Red Cedar River is a 51-mile river with a mixed-use watershed area of approximately 461 square miles. Parts of the river are considered impaired due to dissolved oxygen concentrations (Michigan State University Red Cedar Management Plan 2015). River water was collected in a sterile Nalgene bottle and stored at 4˚C until needed, but no longer than 24 hours.

## Chemicals

All solutions were prepared with MilliQ purified water using chemicals of molecular biology grade. Glacial acetic acid ReagentPlus and crystal violet were purchased from Sigma-Aldrich. Crystal violet at 0.1% (w/v) was prepared by dissolving the dye in MilliQ™ water and filtering through a 0.2 μm filter (Millipore). Physiological saline (0.85% NaCl, w/v) was autoclaved prior to use. Poly(ethylene glycol) with average molecular weight of 200 was purchased from Sigma Aldrich (P3015) as was Potassium Hydroxide (ACS Reagent Grade) Sodium Pyruvate, Dipotassium Phosphate and Magnesium Sulfate. R2A agar, Yeast Extract, Soluble Starch, Proteose Peptone No.3 and Case Amino Acids were purchased from Difco.

## Formation of biofilm and challenge with river water

Biofilms of isolates and controls were formed in 24-well microtiter plates (Corning Costar® 3526) incubated at 25˚C on an orbital shaker at 100 RPM for 48 hours. All plates were sealed with sterile foil covers (VWR Scientific). In single isolate biofilms, 500 μl of an overnight culture grown in R2Broth was added to 1.0 ml of sterile R2Broth in a well. In two-isolate biofilms, 250 μl of each culture was added to 1.0 ml of sterile R2Broth. Our extensive experience with the bacterial strains in this investigation indicated that overnight cultures (16 hrs) used as inocula, consistently produced strain-dependent amounts of biofilm at 24 and 48 hours. This was statistically confirmed in each experiment with four replicates for each treatment. For measuring biofilm biomass, each unique 48-hour biofilm had four replicates. For determination of the phylogenetic structure of communities, each unique biofilm had 4 replicates at each of 3 time points, 4 hours, 8 hours and 24 hours. Microtiter plates were measured at 600 nm using a Biotek Epoch plate reader to confirm growth during incubation. Biofilm screening assays of over 600 freshwater isolates indicated that the amount of biofilm produced under our experimental conditions (R2broth, 25˚C shaking in microtiter plates) was strictly strain-dependent (D. Ye, T.L. Marsh, in preparation). Indicative of this observation was the diversity of biofilm biomass produced by the seven strains included in this study. Each strain produced a unique amount of biofilm in 48 hours as measured by a traditional crystal violet assay and this 48-hour biofilm was what was used in the challenge experiments. In addition, the larger well format afforded by the 24 well plates eliminated concerns regarding air-liquid surface films

and damage to biofilms during washing and staining. In previous studies, air-liquid surface biofilms were problematic when the amount of biofilm production was optimized in select strains by the addition of exogenous protein (44). Only solid surface-liquid biofilms were observed under our experimental conditions described here and no ripping or tearing of biofilm was observed.

Established biofilms of each isolate were challenged with river water in two experiments. The first experiment was designed to determine the effect of filtered and unfiltered river water on established biofilms. Biofilms developed over 48 hours of incubation were washed twice with 1.5 ml of sterile Physiological Saline (PS) and then incubated for an additional 24 hours in the presence of 1.5 ml of either filtered (filtered twice through 0.22 μm Millipore filter) or unfiltered river water. Incubation was at 25˚C on an orbital shaker at 100 RPM. Biofilm was then measured by the crystal violet staining procedure as described below. The second experiment was designed to determine the influence of established biofilm on colonization by river populations. Established 48-hour isolate biofilms were washed twice with sterile PS and then incubated with 1.5 ml of river water for 4-, 8- and 24-hours and then destructively sampled for community analysis. The positive control for all experiments was *P. aeruginosa* PAO1, a robust biofilm forming strain, including under the experimental conditions used in these experiments. The formation of river biofilm in the absence of any pre-established isolate biofilm provided a null control, against which communities derived from the established isolate biofilms were compared. Any deviations from this null control were interpreted as evidence of intervention in river biofilm assembly by the established biofilm.

## Measuring biofilm biomass with crystal violet

Total biofilm biomass was inferred from the crystal violet assay [49, 50]. Briefly, after incubations, well supernatants were gently removed by pipetting and the biofilms were washed twice with sterile PS. The biofilm was stained with 1.5 ml of 0.1% crystal violet (w/v) per well for 15 minutes shaking at 100 RPM on an orbital shaker at 25˚C. The dye was removed, and the plates were gently washed three times with distilled water and inverted to dry overnight. The biofilm-associated dye was extracted with 1.5 ml of 30% acetic acid (v/v) per well for 15 minutes, at 100 RPM on an orbital shaker at 25˚C. The extracted dye in acetic acid was pipetted into a new microtiter plate and absorbance was measured at 600 nm using a Biotek Epoch plate reader. Each experimental condition had four replicates. Four uninoculated wells with broth served as a negative control, the mean of which was subtracted from the absorbance of each experimental well.

## Extraction of DNA for community analysis

After biofilms were established and challenged with river water, plates were removed from incubation and washed twice with 1.5 ml of PS. Cells within the biofilm were lysed by the addition of 1.2 mls of alkaline PEG (pH = 13.3) and incubation at 25˚C, shaking at 100 RPM, for 15 minutes [51]. The plates were sealed with sterile foil and stored at -20˚C until needed for PCR amplification. To determine the phylogenetic structure of the river community, 500 ml of freshly sampled river water were filtered through a 0.22 μm filter (Steritech) with a gentle vacuum assist. This determination was performed in triplicate. The filters bearing the retentate were transferred to 50 ml Corning ® centrifuge tubes containing 80% (v/v) ethanol and stored at 4˚C. Just prior to extraction, the filters and solutions were vortexed vigorously and the cells were pelleted by centrifugation (20 minutes in a Sorvall Fiberlite F18 rotor at 12,000 RPM). After decanting, the pellets were extracted with 100 μl of alkaline PEG [51].

## Illumina sequencing

Illumina sequencing was performed at the Michigan State University Research Technology Support Facility (RTSF). 1–3 μl of alkaline-PEG extracted DNA was used to amplify the V4 region with primers dual indexed fusion primers targeting the V4 region (515f/806r) of 16S rRNA genes. The primers without barcodes and linkers were; forward, 5'-GTGCCAGCM GCCGCGGTAA-3'; reverse, 5'-GGACTACHVGGGTWTCTAAT-3' [52]. Amplification products were normalized (Invitrogen SequalPrep normalization plate) and cleaned with AMPure XP beads. The pool was then loaded on an Illumina MiSeq v2 flow cell and sequenced with a 500-cycle v2 reagent kit to generate 250 bp paired end reads. Base calling was performed by Illumina Real Time Analysis Software (RTA) v1.18.54 and the output of RTA was demultiplexed and converted to FastQ files with Illumina Bcl2fastq v1.8.4. Sequence reads have been deposited in the NCBI Sequence Read Archive (BioProject ID PRJNA849629).

## Sequence analysis and statistics

Sequences from Illumina were processed with Mothur v1.44.1 [53] using the Silva 16S rRNA database Release 138 [54] and Training set 18 of the Ribosomal Database Project [55]. In Mothur, the make.contig command aligns sequence pairs and identifies any mismatched sites and these sites are resolved using the quality scores. For example, where one strand has a gap and the other a base, the base is considered valid only when the quality score of the base is above 25. If there are two different bases at a position, then the base with a quality score six greater or more is taken as the base call. If the quality score difference is less than six the consensus base is set to N. The resulting contigs are then screened and all sequences with ambiguous bases or longer than the expected length are removed. The resulting collection of sequences are trimmed to similar lengths. Mothur uses the chimera.vsearch command to identify and remove chimeras. This program is in the public domain and available at Github. All samples were rarefied to 11,100 sequences. OTU identification was at the 97% similarity level of 16S rRNA using the RDP Training set. This cutoff defines what we refer to as an "OTU" or "population", used interchangeably herein. The calculation of ecological diversity indices of bacterial communities, cluster analysis (UPGMA-Bray-Curtis), ordination (NMDS-Bray-Curtis), ANOSIM [56] and PERMANOVA [57] (both with Bray-Curtis matrix) and SIMPER [56] were performed in PAST 3.0 (Paleontological Statistics Software Package For Education and Data Analysis [58] and based on 10,000 OTUs. Cluster dendrograms generated in PAST were edited in FigTree v1.4.3. All isolates in this study were run individually through Illumina sequencing and analysis in Mothur, along with the communities. This permitted the phylogenetic identification of these isolates within the context of the community analyses and was an important step, in that it allowed us to subtract, before comparative analyses, the sequence signal of the populations that established the initial 48-hour biofilms, the "founder populations", from the communities that evolved after exposure to river water. This provided a more sensitive assessment of the differences between the detected OTUs forming the river biofilm and those that were incorporated into established biofilms. In addition, the phylogenetic signal of our isolates permitted the identification of these populations in the river community. The t-test (assuming unequal variances) as implemented in Excel 16.4 was used to determine the significance of differences in crystal violet-stained biofilm biomass in Fig 2.

## Results

Our objective was to determine if initial colonization of a surface by one or two isolates from the sturgeon egg, altered subsequent colonization of the surface by river bacterial populations. This was driven in part by previous observations that some populations produced

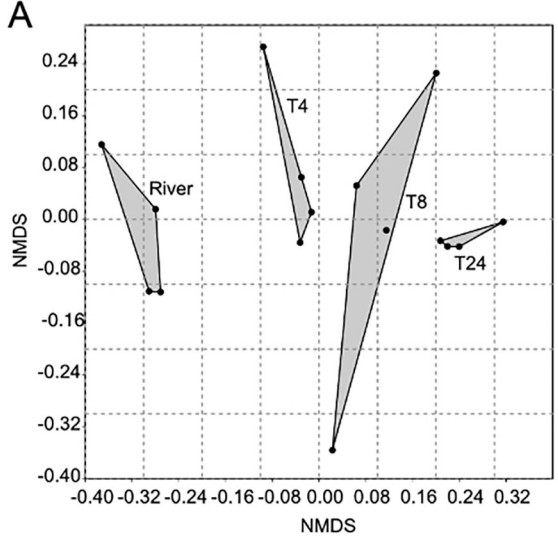

**Fig 1. Ordination, statistical analyses and diversity measures of the river and river biofilm communities.** A. Nonmetric multidimensional scaling analysis of replicates from river and river biofilms at 4, 8 and 24 hours (T4, T8 and T24). B. Pairwise ANOSIM (left block) and PERMANOVA (right block) analyses of the river and biofilm communities at 4, 8 and 24 hours (T4, T8 and T24). Note that the R and F values are black text on white background and corresponding P values are white text on black background. C. Diversity measures of river and biofilm communities. Diversity measurements were calculated from the sum of 4 replicates for each sample. Taxa_S is the total number of taxa detected, Simpson 1_D is the inverse of the standard Simson diversity measure and Shannon H is the Shannon-Wiener index.

antimicrobial compounds [41] and that intervention early in the assembly of the egg-associated microbiome could influence egg mortality [25, 27]. It was important therefore, to determine the assembly pathway of river biofilm in the absence of our test populations. MiSeq 16S targeted sequencing was used to determine the community composition during the assembly of biofilm from river populations. Samples of biofilm were taken at 4, 8 and 24 hours. The differences between the river and river biofilm communities that developed on microtiter plates were statistically evaluated with NMDS, ANOSIM and PERMANOVA and are presented in **Fig 1**. The NMDS plot (Panel A) was cast with three dimensions and only coordinates 1 and 2 are plotted. The stress values for 2 and 3 dimensional plots were 0.1081 and 0.0683, respectively, indicating good to excellent ordination with little or no chance of misinterpretation.

Panel B contains analyses of community data with ANOSIM and PERMANOVA. Again, clear differences between the river and biofilm communities were indicated by ANOSIM R values and PERMANOVA F values. We interpret ANOSIM R values of 0.75 or greater to be a strong rejection of the null hypothesis that there are no substantial differences between the compared conditions [56]. Similarly, PERMANOVA F values greater than 3.0 invalidate the null hypothesis [57, 59]. The river community is statistically different from all three biofilm time points while the 4-hour and 8-hour biofilm communities were the most similar and the 24-hour biofilm appeared statistically different from the 4- and 8-hour biofilms. Panel C presents the diversity measures for the river and biofilm communities. Both the detected (Taxa_S) and the estimated (Chao-1) number of taxa were higher in the river samples than the biofilms. The Simpson and Shannon diversity indices show that diversity was greatest in the river and lowest in the 24-hour river biofilm. The early biofilm timepoints (T4 and T8) were moderately more even than river water and substantially more even than the biofilm at 24 hours.

In Panel A of **Fig 2** the top 25 most abundant OTUs detected in Red Cedar River are shown. These account for 52% of all sequences. In Fig 2, note that the numbers in parentheses following the taxonomic rank indicate the confidence level of phylogenetic assignment. To highlight the shifts in community structures, the columns labeled 4, 8 and 24 hours indicate the rank abundance of these river OTUs in the developing biofilm at the indicated times. Twelve of the top 25 river OTUs were among the top 25 OTUs found in biofilm at four hours. This dropped to six at eight hours and finally five in the 24-hour biofilm. The top 25 OTUs identified in 24-hour biofilm account for 84% of the sequences. Most river populations in the top 25 experienced a rapid drop in relative abundance in the developing biofilm. For example,

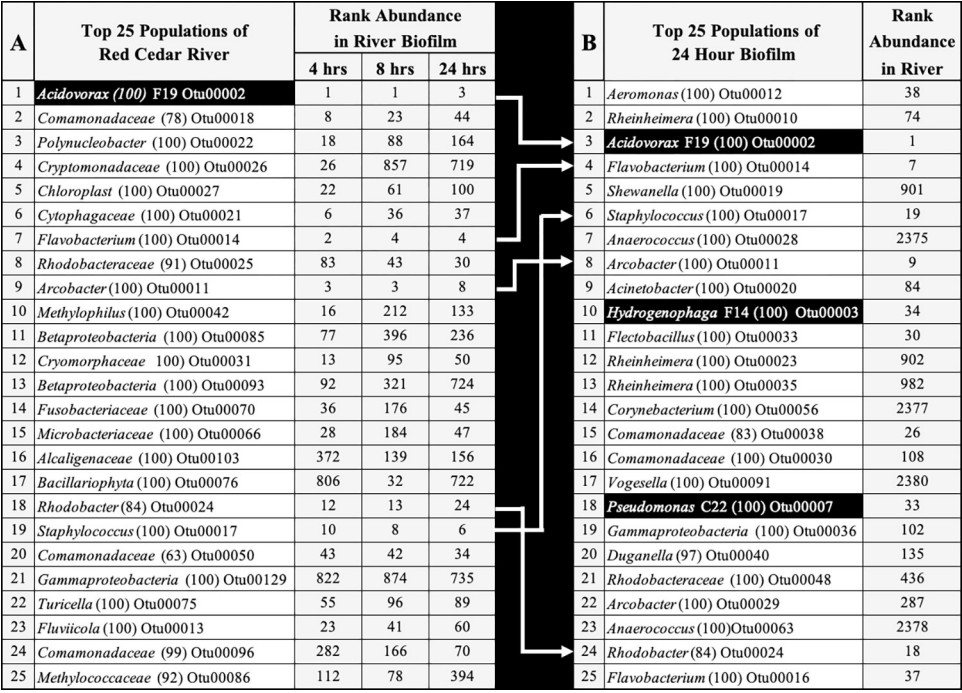

**Fig 2. Top 25 river and river biofilm populations.** Panel A lists the top 25 bacterial populations detected in the Red Cedar River (columns 1 and 2). The three columns to the right report the rank abundance of these populations in river biofilm at 4, 8 and 24 hrs. Panel B lists the top 25 bacterial populations detected in the 24-hour river biofilm. The column to the right reports on the rank abundance of these populations in river water. The white arrows identify the five populations that are shared in river water and 24-hour river biofilm. The four populations shaded black were previously isolated from the surface of sturgeon eggs.

the archetypal freshwater populations of *Polynucleobacter* and *Cryptomonadaceae*, third and fourth in rank abundance in river water, dropped to 164 and 719 rank abundance, respectively, in 24-hour biofilm. Panel B presents the top 25 OTUs in the 24-hour biofilm. The column to the right indicates the rank abundance of these populations in river water. The five persistent river OTUs from the top 25 that were retained in the 24-hour biofilm are indicated by the white arrows and include *Acidovorax* F19, *Flavobacterium* sp., *Arcobacter* sp., *Rhodobacter* sp. and *Staphylococcus* sp. Only three populations of the top 25 in the 24-hour biofilm (*Flavobacterium* OUT00014, *Staphlococcus* OTU00017 and *Hydrogenophaga* OTU00003) had strong positive correlations with their population sizes in river water. Three populations of 24-hour biofilm, identified as *Acidovorax* F19, *Hydrogenophaga* F14 and *Pseudomonas* C22 (shaded cells) were previously isolated from sturgeon eggs [26]. Noteworthy is the observation that some populations that appeared in the top 25 of the 24-hour biofilm were extraordinarily low in rank abundance in river water, rising from a ranking of 2000+ in some cases. The top 30 populations of river biofilm at 4-, 8- and 24-hours in comparison with the river community is shown is S1 Fig. The Taxa_S, Chao 1 and Evenness measures are presented in S2 Fig.

As mentioned above, we were investigating interactions between populations during the assembly of the egg-associated biofilm. To that end we had previously isolated and characterized over 100 isolates from the sturgeon egg [41]. In the experiments described below, we determined the effect that an established biofilm, comprised of one or two isolates, had on the recruitment of river populations into the biofilm matrix. From the strain collection isolated from sturgeon eggs, two Gram positives, *Bacillus* C20 and *Deinococcus* F4, and four Gram negatives, *Serratia* sp. D14, *Hydrogenophaga* sp. F14, *Brevundimonas* sp. F16 and *Acidovorax* sp. F19 were tested. To initiate these studies, a biofilm assay using crystal violet staining was performed. We established eight single-isolate 48-hour biofilms and two-isolate combinations of *Hydrogenophaga* F14-*Brevundimonas* F16 and *Brevundimonas* F16–*Acidovorax* F19. After the biofilms were established, each was challenged with filtered or unfiltered river water for an additional 24 hours, followed by crystal violet staining. The results are presented in **Fig 3**. Under our experimental conditions (R2broth at 25˚C), *P. aeruginosa* PAO1 produced a robust signal of ~3.7 $A_{600nm}$ at 48 hours in the crystal violet assay, indicating considerable biomass in the biofilm. While we have used *P. aeruginosa* PAO1 as a robust biofilm forming strain, as we point out in the Material and Methods section, strains of *P. aeruginosa* are routinely isolated from freshwater systems. Compare this value with considerably weaker biofilms produced by *Serratia* spp, *Hydrogenophaga* and *Acidovorax* that are 1/10 the value. The remaining isolates and combinations produced a diverse range of absorbances, indicating greater or lesser biofilm, depending on the isolate (black bars). *Deinococcus* and the two-isolate combinations (F16 + F14 and F6 + F19) showed the most robust biofilms at 48 hours. Regarding the challenge with river water, there were three phenotypic responses. First, there were isolates that produce very low amounts of biofilm such that even though there was statistical distinction between the challenged biofilm and its cognate control, it was difficult to draw a strong conclusion. In this group were the two *Serratia* spp., *Bacillus* C20 and *Hydrogenophaga* F14. The second phenotype included *P. aeruginosa* PAO1 and *Deinococcus*. In these two strains, both filtered and unfiltered river water reduce the amount of detectable biofilm compared to the unchallenged 48-hour sample. In the case of *Deinococcus*, the reduction was quite pronounced. The final phenotypic response, exhibited by *Brevundimonas*, *Acidovorax* and the two-isolate combinations of *Brevundimonas* with either *Hydrogenophaga* or *Acidovorax*, was a statistically significant diminution of biofilm when challenged with filtered river water, accompanied by either no change, or an increase in biofilm, when challenged with unfiltered river water. In the case of *Brevundimonas* and the combination of *Brevundimonas* with *Acidovorax*, filtered river water diminished biofilm and unfiltered water substantially increased biofilm amounts.

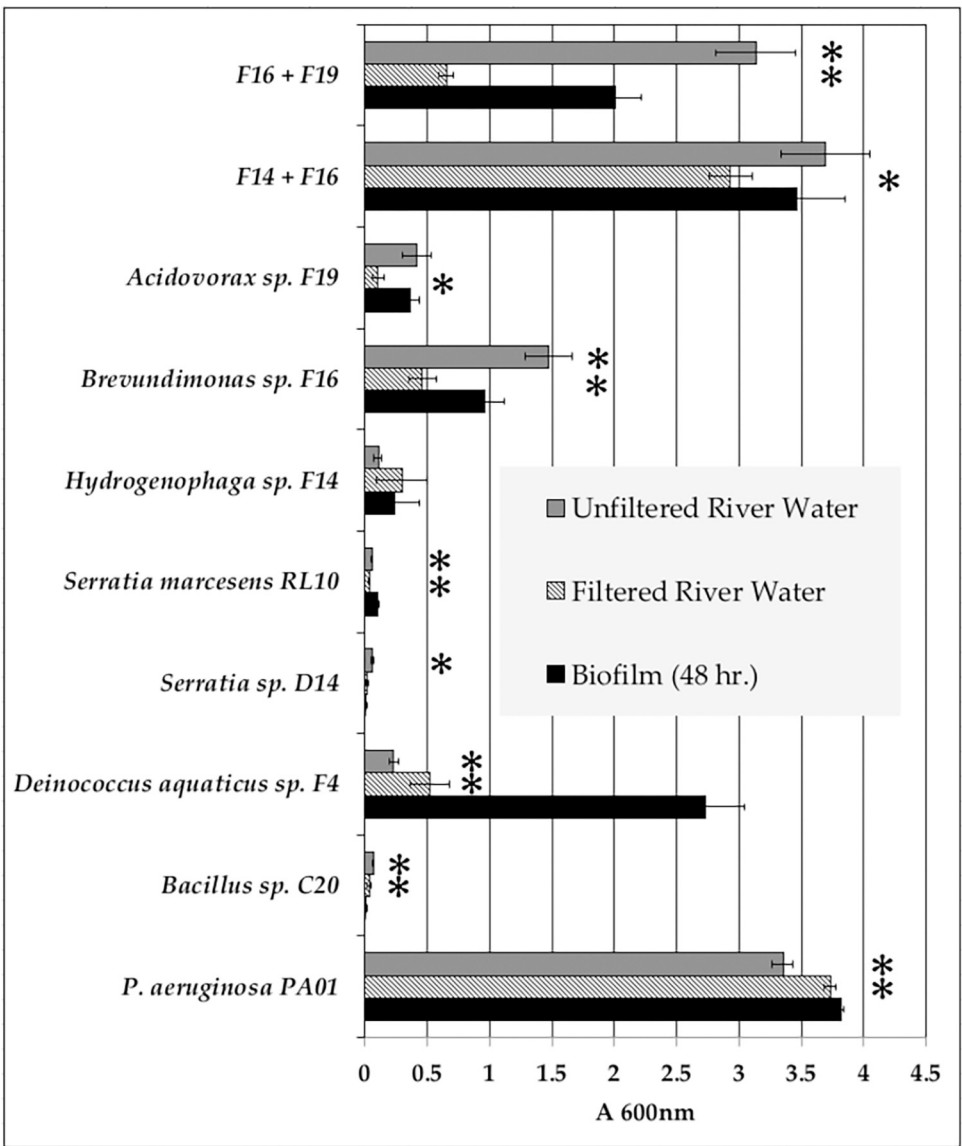

**Fig 3. Resilience of single and double isolate biofilms to filtered and unfiltered river water as measured with crystal violet.** The plotted 600 nm absorbance values (abscissa) are the mean of four replicate biofilms of each of the tested strains (ordinate) and standard deviations are indicated with bars. Striped and gray bars represent biofilm challenged with filtered and unfiltered river water, respectively. Asterisks indicate responses that were statistically different (P value = 0.05 or less) from the cognate control 48-hour biofilm (black bars).

To determine which river populations were accepted or excluded from the established biofilms of the isolates, experiments similar to those described above were set up where river water challenged 48-hour biofilm of the eight isolates and the two combinations. Samples were taken at 4, 8 and 24 hours, the biofilm DNA was extracted and 16S rRNA targeted Illumina sequencing determined the phylogenetic composition of the biofilms. The resilience of the established biofilms to invasion by wild populations is presented in **Fig 4** where the percent of the founding population(s) remaining in the biofilm is plotted as a function of time. Three groups can be identified based on the percent of total sequences at 24 hours. There were four strains or strain combinations that maintained 80% or greater of the community, based on

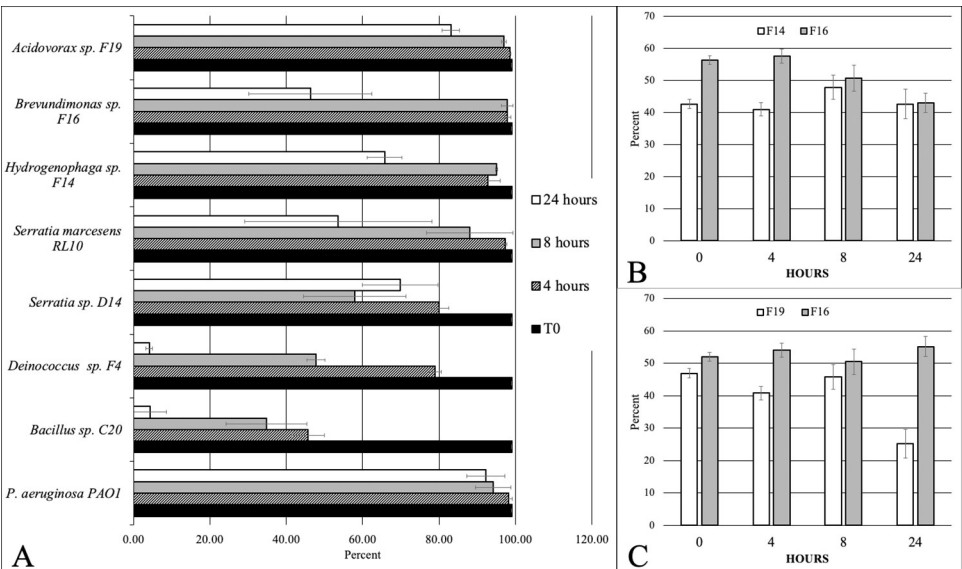

**Fig 4. Resilience of established biofilm to challenge with river bacterial populations as determined by 16S amplicon sequencing.** Panel A, percent (mean of four replicates) of the founding populations remaining in biofilms at 4, 8 and 24 hours. Panels B (F14 & F16 mixed biofilm) and C (F19 & F16 mixed biofilm), percent (mean of four replicates) of the founding two-isolate biofilms remaining after 4, 8 and 24 hours. Percent is calculated based on the entire community (11,100 sequences), including isolates. Error bars display the standard deviation based on four replicates.

total sequence count, *P. aeruginosa* PAO1 (92%), *Acidovorax* F19 (82.1%), and the strain combinations of *Brevundimonas* and either *Hydrogenophaga* F14 (85.6%) or *Acidovorax* F19 (80.3%). The second grouping included isolates that maintain between 46% and 70% of the community. These included *Serratia* spp. D14 (69.8%) and RL10 (53.6%), *Hydrogenophaga* F14 (65.7%) and *Brevundimonas* F16 (46.3%). Finally, the third group included the two founding populations of *Bacillus* C20 and *Deinococcus* F4 that were reduced to minor population status at 4.28% and 4.08%, respectively. Noteworthy is the observation that in the two-isolate combinations of *Brevundimonas* with either *Hydrogenophaga* or *Acidovorax*, both strains of the combinations occupied roughly an equivalent fraction of the community (45%-55%).

Statistical analyses were performed on the communities to determine if prior colonization by one of the isolates predisposed the surface to a unique biofilm assembled from river populations. **Fig 5** presents the cluster analysis of the ten isolate communities and the river biofilm at the three time points. The founding populations were included in the top row of these analyses and excluded from those of the bottom row. Because the founding populations are large, particularly at 4 and 8 hours, they drive the clustering of the top row. Note the tightly clustered replicate communities at the end of long branches at 4 hours. As the established biofilms begin to deteriorate and river populations assume greater dominance in the biofilm community, the tightness of replicate clustering diminishes. In the bottom row, while the replicate clustering was, by in large retained, there is greater volatility in clustering, especially at 8 and 24 hours among the least stable biofilms. For example, *Bacillus* C20 and *Deinococcus* F4 were the isolates with the least resilient biofilms and the clustering of replicates from these communities split by 8 hours and remained so at 24 hours. Those biofilms from isolates with an intermediate level of resilience were mixed with regards to clustering stability. *Brevundimonas* F16 replicates remained clustered while three of four replicate communities of *Hydrogenophaga* F14 were clustered at 24 hours. Replicate clusters of the two *Serratia* isolates deteriorated by 24 hours.

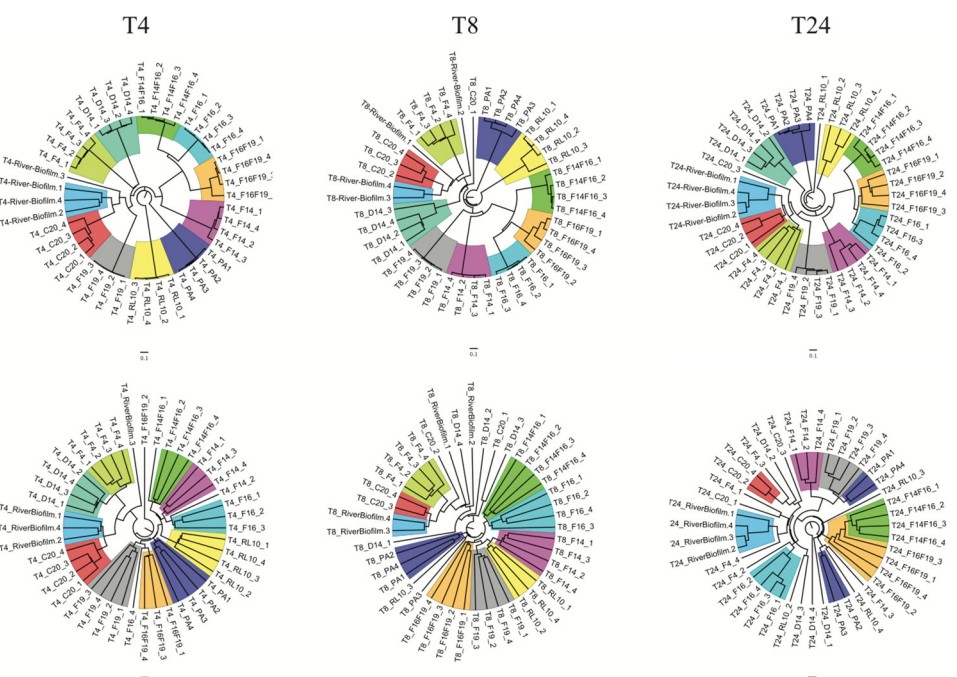

**Fig 5. Cluster analysis of biofilms communities at 4, 8 and 24 hours (T4, T8, T24).** Cluster analyses (UPGMA-Bray-Curtis) were performed with (upper row) and without (lower row) the founding populations. Clusters of founding populations are color-coded systematically in all dendrograms to facilitate tracking.

Those communities where the founding population remained at 80% or greater after 24 hours, retained their clustering, with the exception of PAO1communities that become split.

As mentioned above, the isolates *Hydrogenophaga*, *Brevundimonas* and *Acidovorax* were of particular interest to us because of previous investigations. The NMDS, ANOSIM and PER-MANOVA analyses of the 24-hour biofilm communities of these isolates are presented in **Fig 6** and show that the isolates and isolate combinations were statistically different. These analyses were performed on datasets from which the founding populations (the initial biofilm) had been subtracted. Clear separations of the communities were seen on NMDS plots with stress values that suggest unambiguous ordination. ANOSIM R values and PERMANOVA F values all indicated that there were statistically significant differences between all communities, including between the two-isolate combinations and the single-isolate cognates. This separation could be seen in the early samples at 4 and 8 hours as well (S4 and S5 Figs). Moreover, as described above, the two-isolate combinations were more resilient to invading river populations than single-isolate biofilms. The top 30 populations for all samples are presented in the supplemental material.

SIMPER analysis is typically used to identify the populations that contribute most to the statistical difference between two communities. All the isolate biofilm communities were compared with the river biofilm by SIMPER, to determine how biofilm of these isolates influenced the assembly of biofilm from river populations. The top 50 populations identified by SIMPER are presented in **Fig 7** where the established biofilms are ordered from left to right by increasing resilience to invading river populations (see Fig 3). All OTUs presented in this Figure were identified as contributing 0.5% dissimilarity or greater and had a minimum of 200 sequences. SIMPER analysis allowed us to identify populations that were i.) blocked from establishing a sizable presence in the established biofilm (black filled), ii.) invasive to the established biofilm

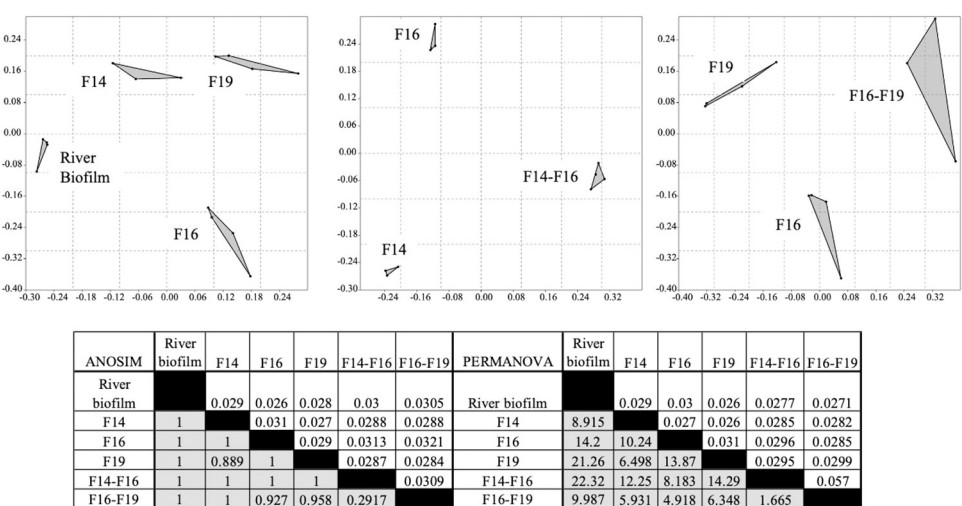

| ANOSIM | River biofilm | F14 | F16 | F19 | F14-F16 | F16-F19 | PERMANOVA | River biofilm | F14 | F16 | F19 | F14-F16 | F16-F19 |
|---|---|---|---|---|---|---|---|---|---|---|---|---|---|
| River biofilm | | 0.029 | 0.026 | 0.028 | 0.03 | 0.0305 | River biofilm | | 0.029 | 0.03 | 0.026 | 0.0277 | 0.0271 |
| F14 | 1 | | 0.031 | 0.027 | 0.0288 | 0.0288 | F14 | 8.915 | | 0.027 | 0.026 | 0.0285 | 0.0282 |
| F16 | 1 | 1 | | 0.029 | 0.0313 | 0.0321 | F16 | 14.2 | 10.24 | | 0.031 | 0.0296 | 0.0285 |
| F19 | 1 | 0.889 | 1 | | 0.0287 | 0.0284 | F19 | 21.26 | 6.498 | 13.87 | | 0.0295 | 0.0299 |
| F14-F16 | 1 | 1 | 1 | 1 | | 0.0309 | F14-F16 | 22.32 | 12.25 | 8.183 | 14.29 | | 0.057 |
| F16-F19 | 1 | 1 | 0.927 | 0.958 | 0.2917 | | F16-F19 | 9.987 | 5.931 | 4.918 | 6.348 | 1.665 | |

**Fig 6. NMDS, ANOSIM and PERMANOVA analyses of biofilm communities from river and pre-established biofilms of *Hydrogenophaga* F14, *Brevundimonas* F16, and *Acidovorax* F19 at 24 hours incubation.** Sequences of the founding populations have been removed prior to comparative analyses. R values and F values are shaded and uncorrected P values are unshaded.

(red filled), or iii.) selected for by the established biofilm (green filled). In this figure a blocked population (black cells) was one of the major populations of river biofilm that was reduced at least 10-fold and frequently to single digit sequence representation in isolate biofilm communities comprised of 11,100 sequences. Those populations that appeared successfully invasive (red cells) to established isolate biofilms were frequently comparable in number to what was found in unchallenged river biofilm. Finally, those populations that appeared selected for (green cells) had a robust presence in the established biofilm but little or no presence in river biofilm. All the established biofilms blocked several of the listed populations and *Anaerococcus*, *Arcobacter* and *Staphylococcus* were blocked by all of the established biofilms. *Aeromonas* and *Rheinheimera* were the two most invasive populations that infiltrated five and eight of the biofilm communities, respectively. Only three populations had multiple instances of selection, *Flavobacterium* (3/10), *Fluviicola* (5/10) and *Vibrio* (3/10). All other instances of selection involved a single event and many of these populations were very low in the rank abundance of river biofilm populations (eg. *Bacteriodetes* and *Geobacillus*). *P. aeruginosa* PAO1, the most resilient of the biofilms, had four invasive populations and blocked 20 from invading. Of note, the two-isolate combinations were exceptionally resistant to invasion. *Brevundimonas* F16 with *Hydrogenophaga* F14 blocked 20 river biofilm populations and selected for three, while *Brevundimonas* in combination with *Acidovorax* F19, blocked 21 and selected for five populations. Interestingly, the lone "invasive" population of the *Brevundimonas-Acidovorax* collaboration was *Hydrogenophaga*. *Aeromonas*, a genus with fish pathogens, was blocked by 5 of the biofilms, including the two *Serratia* isolates and both two-isolate biofilms. The blocked populations were ranked 1–28 in the river biofilm whereas the invasive populations were ranked 1–21. Populations that were selected by the established biofilms were ranked from 26–2000 + in river biofilm.

## Discussion

The assembly of microbial biofilms is a reasonably well-studied phenomenon [61–65], although given the diversity of microbes and habitats, we do not have an expansive view. For

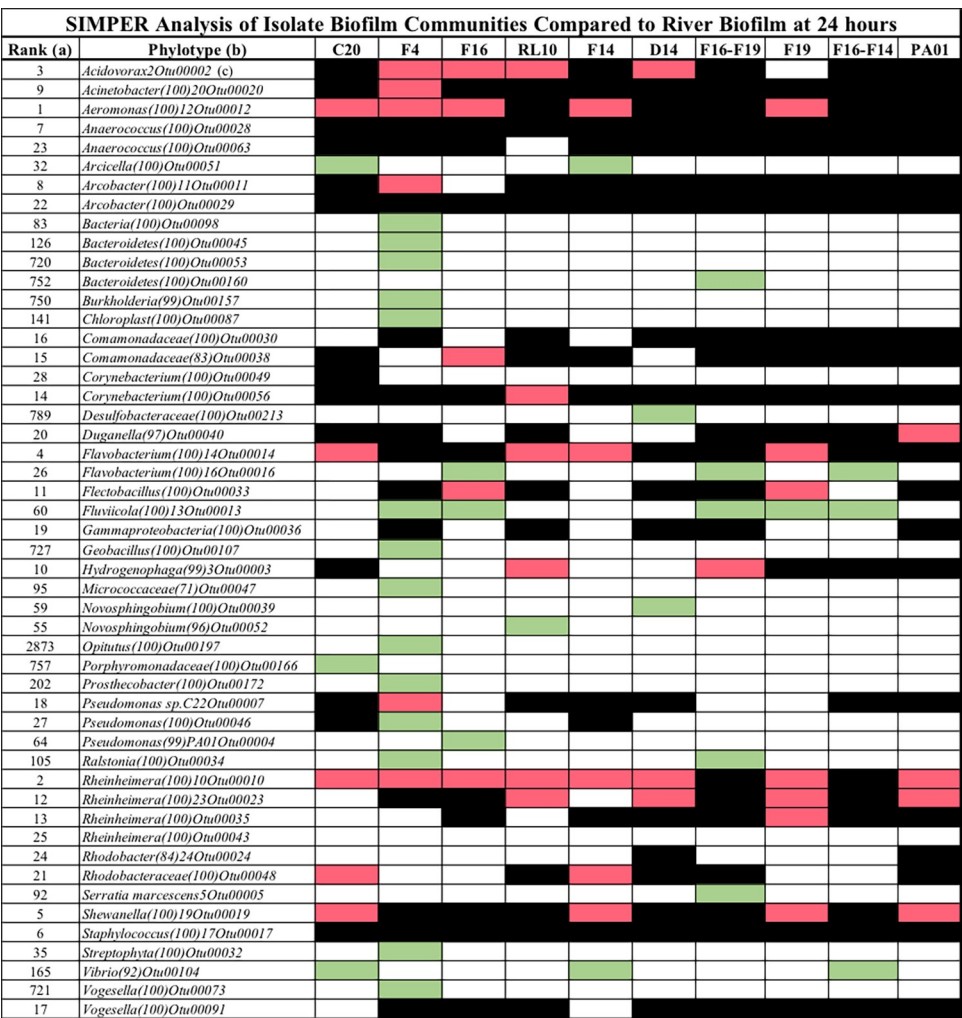

**Fig 7. The fate of river populations when colonizing established biofilms.** All communities from the established biofilms were compared against the river biofilm community with SIMPER (as implemented in PAST) to determine the influence the established biofilms had on community assembly. Black indicates that the population was blocked from achieving numbers comparable to the river biofilm. Green indicates that the population was selected by the established biofilm and found in substantially greater numbers than in river biofilm and Red indicates that the population size was comparable between the established and river biofilms. Only populations that accounted for 0.5% or greater dissimilarity as indicated by SIMPER with a minimum of 200 identified sequences were counted. A minimum of a 10-fold difference in population counts between the river and established biofilms was required for the blocked and selected categories. (a) Rank abundance in river biofilm, (b) OTU as determined by Mothur [53, 60], (c) number in parentheses indicates confidence of phylogenetic assignment. The OTU number is assigned by Mothur and used to track OTUs.

example, we do not know if all free-living species of bacteria can form biofilms, but it has become apparent that a substantial fraction of bacteria in the environment exists as biofilm [66]. Moreover, the assembly of mixed species biofilms occurs through a temporal sequence of species additions [62, 64], suggesting that primary colonists may establish a pathway for subsequent colonization. In the case of river biofilm communities, Besemer et al, [61] has shown that the phylogenetic composition of mature biofilm is different and less diverse than the stream community. Moreover, biofilm communities from three different streams were more similar than the stream communities, suggesting strong selective filtering in the assembly of stream biofilms [61]. Several reports identify species interactions as a driving force in biofilm

assembly [67–69] and unique emergent properties have been identified in mixed species bio-films [70, 71].

As mentioned above, our interest has been on the impact of the sturgeon egg microbiome on survival and mortality of the egg. The egg initially presents as a tabula rasa, devoid of a microbiome, but within minutes of exposure to stream water, there is a detectable and diverse microbiome [26, 27]. We view this microbiome assembly as a three-stage process where during the first period, 0–8 hours after release from the female, aquatic microbial populations with chemical compatibility with the surface of the egg, bind and become established. At this early point in the microbiome assembly there are few ecological pressures on colonizing populations other than the ability to bind to the egg's surface with sufficient strength to avoid wash off from the sheer of water turbulence. Perhaps for this reason we see greater diversity during the early stages of microbiome assembly. During the second period, up to 24 hours post release and fertilization, there is a large shift in community structure compared to the early period. We posit that during the first 24 hours the early colonizers become subjected to host defense mechanisms [72–74], interspecies competitions [75–79] and predation from bacteriophage and bacteriorvores [80–84], and that these pressures substantially modify the community. During the next 5–6 days, the community of a healthy egg experiences small community shifts and a slow increase in the size of the community on hatchery-reared eggs [25, 27]. Measuring the same kinetics in eggs deposited in streams is considerably more difficult.

The assembly of the river biofilm community on sterile polystyrene plates is similar to community compositional development on uncolonized lake sturgeon eggs following extrusion by the female during spawning perhaps only in the sense that both surfaces are previously uncolonized. When colonizing polystyrene, bacterial community values for diversity, number of taxa, and Chao 1 were all greater at 4 and 8 hours compared to 24 hours. The evenness of populations was also greater at 4 and 8 hours compared to the 24-hour biofilm and the river community. Moreover, the areas defined by convex hulls of the NMDS plots were greater at 4 and 8 hours compared to 24 hours, indicating greater variability in the four replicates at early time points. These data reveal more diverse, even, and volatile early communities moving towards more select, skewed, and less phylogenetically variable communities by 24 hours. Both ANOSIM and PERMANOVA indicate that dissimilarity between river water and river biofilm is in the order 24 hrs > 8 hrs > 4 hrs, with good statistical support. Of the top 25 OTUs from the river biofilm at 24 hours, 13 were from populations that were in the top 100 most abundant river populations and 8 were ranked less than 400th in abundance of river populations. Thus, in the construction of this biofilm over 24 hours, both relatively numerous and scarce populations were selected, and selection was substantial in that populations with rank abundances of less than 2000 in river water were in the top 25 populations of biofilm. *Aeromonas spp.*, the most abundant genera of 24-hour river biofilm, is a potential fish pathogen. In hatchery-reared eggs of the lake sturgeon we have not detected this abundance of *Aeromonas*. However, in stream captured eggs, *Aeromonas* was the most abundant population (Ye, Scribner & Marsh, unpublished). *Pseudomonas* C22_Otu00007, the 18[th] most abundant population in river bio-film, was isolated from sturgeon eggs [41] and through comparative genome analysis (Angoshtari, Scribner & Marsh, unpublished), found to be closest to *P. fluorescens* strain 48D1, isolated from soil (Genbank accession # MOBT01000000).

Before establishing the phylogenetic effects that single and double isolate biofilms had on the assembly of biofilm from river populations, we assessed the resilience of isolate biofilms to river water using the crystal violet assay. These data were provocative in that some biofilms (eg. *Deinococcus* F4) deteriorated when exposed to either filtered or unfiltered river water but several biofilms (eg. *Brevundimonas* F16, *Acidovorax* F19 and F16 in combination with either *Hydrogenophaga* F14 or F19) were diminished when challenged with filtered water but

enhanced when exposed to unfiltered river water. Simple interpretations of these data are that biofilm fragility or durability is isolate specific, and that river populations enhanced the survival of some isolates within biofilm, but not all. One possible explanation for this enhancement is via synergistic interactions between species. In previous work we detected increased biofilm formation in the isolate combinations of F16 with either F14 or F19 when challenged with tobramycin [42]. Here we see increased biofilm and resistance to invasion by river populations in the two-isolate combinations. This suggests the development of an emergent property as a result of interactions between isolates within the biofilm. Noteworthy were the sequencing results that indicated roughly equal representation of the two isolates in the combination biofilms.

While clearly our interest lies in the assembly of the microbiome on sturgeon eggs, we elected to run our experiments within the confines of a well-controlled polystyrene microtiter plate. These plates provided a convenient platform with which to replicate, establish and challenge biofilm. To the bacterium, the microtiter plate presents as a benign, relatively smooth surface that is negatively charged and hydrophilic with 9–17% oxygen atoms (Corning Life Sciences Product Description). When extrapolating to the egg, we must keep in mind the many complexities that are absent, for example the dynamics of a fertilized egg with maternally provisioned defenses [72–74, 85–89] and a metabolizing embryo within, that is consuming oxygen and excreting nitrogenous waste and $CO_2$. In addition, the egg's surface is highly cavitated, providing many surface irregularities for bacteria to attach [41]. Nonetheless, we can glean several attributes of our isolates that strongly suggest further investigations in the hatchery. Of note are the populations that appear to inhibit colonization of fish pathogens. Four biofilms were particularly resistant to invasion by river populations including *P. aeruginosa* PAO1, *Acidovorax* F19 and two-isolate combinations of *Brevundimonas* F16 with either *Hydrogenophaga* F14 or *Acidovorax* F19. Our positive control, *P. aeruginosa*, displayed a robust and resilient biofilm, impervious to invasion over 24 hours. We have detected putative populations of *P. aeruginosa* on eggs at early timepoints (15–90 minutes) but these are lost by 24 hours (Angoshtari & Marsh, in preparation) and *P. aeruginosa* is found widely distributed in soil and streams [45]. *Acidovorax* is a Gram negative non-sporulating rod with aerobic respiration. It has been isolated from soil [90], plant roots [91], freshwater and marine systems [92], and from Sturgeon eggs [41] from the Black River in northern Michigan. It was the numerically dominant population as measured with Illumina sequencing in the Red Cedar River in southern Michigan (this study). Established biofilm of *Acidovorax* F19 successfully blocked invasion by *Acinetobacter*, *Anaerococcus*, *Arcobacter*, *Corynebacterium*, *Duganella*, *Hydrogenophaga*, *Staphylococcus* and *Vogesella*. *Brevundimonas* is a Gram negative alphaproteobacterial rod with a terminal holdfast, similar in function to the holdfast in the closely related *Caulobacter* genus. Both are commonly found in freshwater systems along with *Hydrogenophaga*, a Gram negative betaproteobacteria. No fish diseases have been linked to these bacteria. The two-isolate biofilms of *Brevundimonas* combined with either *Hydrogenophaga* or *Acidovorax*, had more extensive "blocking" profiles than the isolates alone, including the exclusion of *Aeromonas*, a *Flavobacterium*, *Rheinheimera* and *Shewanella*. Statistically, the blocking and selection of populations by established biofilms appears robust. However, in the case of *P. aeruginosa* PAO1, the founding population represents 92% of the community after 24 hours. With such a robust and resilient biofilm, the invading populations represent 8%, or only 800 out of 11,100 sequences in the community after 24 hours (less at 4 and 8 hours). We suspect that the consequence of these small community fractions was the variability in clustering of the communities derived from this robust biofilm forming strain.

These results suggest the possibility of controlling infections of eggs and larvae through the establishment of a mixed species biofilm early in development. However, while *Acinetobacter*,

*Arcobacter*, *Corynebacterium*, *Staphylococcus*, *Aeromonas* and *Flavobacterium* have known fish pathogens within the genus [93–105] and were blocked in biofilm formation by several of our isolates, *Rheinheimera* and *Shewanella*, potential probiotics [106–111], are also blocked. If we hope to control community assembly on the egg to diminish the inclusion of pathogens, these results underscore the need for a careful assessment of each putative probiotic, accounting for both beneficial as well as detrimental influences on microbiome assembly. Nonetheless, our findings suggest that a multispecies probiotic applied to young eggs and larvae can block the establishment of pathogens within the developing microbiomes. For example, blocking the early establishment of virulent strains of *Aeromonas* and/or *Flavobacterium* on the egg by pretreatment with *Acidovorax*, *Hydrogenophaga* and *Brevundimonas* may improve survival. We have repeatedly observed that intervention early in the development of sturgeon eggs can reduce mortality [25, 27] and early intervention as a method to direct microbiomes towards more optimal structures has been proposed for several systems [112–115]. In hatcheries where eggs can be harvested aseptically from the female, the early establishment of a beneficial microbiome could co-occur with fertilization and potentially reduce the use of environmentally problematic antibacterial treatments. We view the rapid laboratory assay that we employed as a first step in sorting through the myriad of metabolic and competitive interactions between populations in a phylogenetically complex microbiome. A surprising observation was the "rescue" of populations with low abundance ranking in river water that were incorporated into the developing biofilm in substantial numbers. For example, *Deinococcus* F4 appeared to select for populations of Bacteroidetes, *Burkholderia*, *Geobacillus* and *Opitutus* with river water rank abundances of 720, 750, 727 and 2873, respectively. The F16-F19 mixed biofilm selected for a Bacteroidetes and *Serratia* D14 selected for a Desulfobacteriaceae population, both with low abundance in river water. These relative population numbers were evaluated after 24 hours of selective pressures and could indicate that the founding population provided a considerably more suitable binding chemistry for certain low abundance populations, or the selection could be the result of strong metabolic entanglement between the populations. Either way this technical approach could accelerate the identification of ecologically interacting bacterial populations within biofilms.

## Supporting information

**S1 Fig. The top 25 populations of river water and river biofilm.** Panels A, C & E are sorted according to the numerical abundance of river water populations. Panels B, D & F are sorted according to the numerical abundance of river biofilm populations. Panels A&B, C&D and E&F are samples at 4, 8 and 24 hours respectively. Populations that were isolated from eggs in previous studies are indicated with arrows.
(ZIP)

**S2 Fig. Taxa S and Chao 1 diversity indices and evenness of river biofilms at 4, 8 and 24 hours in the presence and absence of preestablished isolate biofilms.** These measurements are derived from community analyses from which the founding isolate populations have been subtracted, thereby providing values that are not substantially influenced by the large population size of the established biofilms. Four replicates were used to calculate the mean and standard deviation.
(TIF)

**S3 Fig. ANOSIM and PERMANOVA analysis of river biofilm and isolate biofilm communities at 4, 8 and 24 hours after removal of founding populations.** S3A, 4hrs, S3B, 8hrs, S3C, 24hrs.
(ZIP)

**S4 Fig. NMDS, ANOSIM and PERMANOVA analyses of biofilm communities from river and pre-established biofilms of *Hydrogenophaga* F14, *Brevundimonas* F16, and *Acidovorax* F19 at 4 hours incubation.** Founding populations of the pre-established communities have been removed prior to comparative analyses. R values and F values are shaded, and uncorrected P values are unshaded.
(TIF)

**S5 Fig. NMDS, ANOSIM and PERMANOVA analyses of river biofilm and isolate biofilm communities from river and pre-established biofilms of *Hydrogenophaga* F14, *Brevundimonas* F16, and *Acidovorax* F19 at 8 hours incubation.** Founding populations of the pre-established communities have been removed prior to comparative analyses. R values and F values are shaded, and uncorrected P values are unshaded.
(TIF)

## Acknowledgments

We thank Dr. M. Bagdasarian (MSU) for providing *P. aeruginosa* PAO1.

## Author Contributions

**Conceptualization:** Roshan Angoshtari, Terence L. Marsh.

**Data curation:** Roshan Angoshtari, Terence L. Marsh.

**Formal analysis:** Roshan Angoshtari, Terence L. Marsh.

**Funding acquisition:** Kim T. Scribner, Terence L. Marsh.

**Methodology:** Terence L. Marsh.

**Project administration:** Terence L. Marsh.

**Resources:** Kim T. Scribner, Terence L. Marsh.

**Supervision:** Terence L. Marsh.

**Writing – original draft:** Roshan Angoshtari.

**Writing – review & editing:** Kim T. Scribner, Terence L. Marsh.

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
