## [Decision Letter · Decision Letter 0]

24 Jan 2023

PONE-D-22-31146The impact of primary colonizers on the community composition of river biofilmPLOS ONE

Dear Dr. Marsh,

Thank you for submitting your manuscript to PLOS ONE. After careful consideration, we feel that it has merit but does not fully meet PLOS ONE’s publication criteria as it currently stands. Therefore, we invite you to submit a revised version of the manuscript that addresses the points raised during the review process.

We look forward to receiving your revised manuscript.

Kind regards,

Robert P Smith

Academic Editor

PLOS ONE

Journal Requirements:

   "Funding sources included the Michigan Department of Natural Resources State Wildlife Grants Program T-10-T-5 Study 237026 (https://www.michigan.gov/dnr/buy-and-apply/grants/aq-wl/wildlife562 hab), Center for Water Sciences Water Cube initiative at MSU (https://water.msu.edu/watercube/), and the Department of Microbiology and Molecular Genetics at MSU. In addition, we thank the Department of Microbiology and the College of Natural Science for partial funding including a Thesis Completion grant. We thank Dr. M. Bagdasarian (MSU) for providing Pseudomonas aeruginosa PA01."

  "Funding sources included the Michigan Department of Natural Resources State Wildlife Grants Program T-10-T-5 Study 237026 to KS (https://www.michigan.gov/dnr/buy-and-apply/grants/aq-wl/wildlife-hab), Center for Water Sciences Water Cube initiative at MSU to TLM & KS (https://water.msu.edu/watercube/), and the Department of Microbiology and Molecular Genetics at MSU (TLM). In addition, we thank the Department of Microbiology and the College of Natural Science for partial funding including a Thesis Completion grant (RA). The funders had no role in study design. data collection and analysis, decision to publish, or preparation of the manuscript."

Additional Editor Comments:

Good afternoon Dr. Marsh,

First, I would like to apologize for the length of time that it took to review your manuscript. As we discussed, it took quite a long time to find qualified reviewers. Nevertheless, I now have received two reviews.

Both reviewers appear to have found the manuscript to be technically sound. However, they have some suggestions about justification about experimental approaches, writing and the figures/figures legends that are worth addressing. I look forward to receiving your revised manuscript. If you have any questions, please do not hesitate to contact me.

Rob

Reviewers' comments:

Reviewer's Responses to Questions

**Comments to the Author**

1. Is the manuscript technically sound, and do the data support the conclusions?

Reviewer #1: Partly

Reviewer #2: Yes

2. Has the statistical analysis been performed appropriately and rigorously? 

Reviewer #1: Yes

Reviewer #2: Yes

3. Have the authors made all data underlying the findings in their manuscript fully available?

Reviewer #1: Yes

Reviewer #2: Yes

4. Is the manuscript presented in an intelligible fashion and written in standard English?

Reviewer #1: No

Reviewer #2: Yes

5. Review Comments to the Author

Reviewer #1: General

The authors investigated how the establishment of biofilms from river microbial communities were affected by pre-established biofilms by isolates from sturgeon eggs and vice versa. They conclude that certain taxa could act as probiotics if they are precolonized on sturgeon eggs and then exposed to possibly harmful taxa in the river communities. In general, I think this is a straightforward and interesting study, however there are several improvements that could be made to the writing and presentation of these data, and important methodology is missing.

The justification for using specific isolates should be highlighted more clearly. I do not see any informative on cell counts or growth phase at which these isolates were inoculated, and I’m not sure the absorbance values presented would allow for successful replication of these results. I There are other pieces of information missing from the methods section that I’ve highlighted below that need to be included.

The English is fine, but the scientific writing, especially in the results, could be improved.

The results could be tightened by removing statements that are not informative to the results. For example, Lines 265-266 don’t seem to add much and may be more appropriate for a legend, and Lines 284-288 are redundant of intro and methods. There is much more than can be cut.

The data presented in the results section and the figures/tables are not always clear. For example Table 1 and Figure 1 panel B and others like it. The legends need to be more way informative so the reader can understand them as well, but in general I think the presentation of data in some tables and figures should be re-considered (Table 1, Figure 1c, Figure 4 [way too small, no description of colors?], Figure 6, Figure 7 [too small of text]). In addition I think the results needs to be described more clearly with more informative information presented in the figures. I could not understand the results without spending time trying to figure out what the figures were presenting, and even then, the legends were missing necessary information.

I think the conclusion could be more strongly supported if there was a known egg pathogen to challenge the biofilm rather than inferring certain pathogens from the microbial community composition of the river. Certainly not all Aeromonas sp. are pathogenic to the eggs? However, I see the value in investigating as a microbial community as well because it is more representative of nature conditions.

Specific

Line 55 awkward sentence

Line 82: bd is not defined before this, I assume it is B. dendrobatidis

Line 83: suggestion: remove “in our labs”

Line 86: suggestion: alter language to avoid first person

Line 93: I think “with” is missing

Line 98. Starting with “our experimental strategy…” – consider removing this and everything after or shorten it.

The details of how isolates were identified need to be added. For example, you just say that partial 16S sequencing was used to identify. In addition, the justification for using these strains is lacking. For example, line124-125 states that these were representative of previous characterizations – is this something you can cite? How abundant are these taxa on the eggs? Are egg communities diverse or do they only harbor a few species? Following that at line 126, do you mean that these were of interest related to biofilms and resilience on eggs or just in general?

Llines 153 and on.. Can you provide more quantitative information about the overnight cultures inoculated? Cell count? Absorbance? Were they added at the same concentration?

Line 160 – avoid starting a sentence with a number.

Line 204 is awkward. Consider changing to “…500 cycle v2 reagent kit to generated 250 bp paired end reads.

Line 205 needs work to clarify.

Line 212-214 is awkward the way it is written. For example, the analysis of diversity of bacterial communities is redundant to all the following analyses listed.

Line 212: permanova

Can you provide the details of your mothur analysis? Trimming, filtering, chimera checks?

Isn’t is “t-test?”

Lines 228: “were” instead of was

The first paragraph of the results section doesn’t seem appropriate for the results, but perhaps intro or beginning of discussion.

Line 238: Is there data somewhere to show this or reference?

Is the sentence starting at line 239 needed? Also seems like the next two sentences are redundant of the methods.

Line 245: I don’t think stress is in an indicator of anything being different but rather an indicator of how well the data is represented in that 2 or 3D model.

Panel B in Figure 1 is not easy to interpret. The shaded boxes and non-shaded boxes need to be explained in the legend. According to the way it is now, one could interpret that T4, T8 etc are R values, because it is not clear here these are timepoints.

In figure 1 panel C what does Taxa S mean? Describe these things in the legend.

Line 267, what are 164 and 719? Rank abundance? You mention relative abundance right before this so this is a bit confusing.

Table one is not clear as is. The lettering is confusing and not intuitive.

Line 320 “described”

Lines 297 : an absorbance of 3.7 means nothing without any comparisons, which do not follow. What were the absorbances of the “low amounts” of biofilm? What is considered “low” and why?

The discussion could be streamlined and made me succinct.

Reviewer #2: Comments from the Reviewer of the Angoshtari et al. manuscript ‘The impact of primary colonizers on the community composition of river biofilm’ (PLoS ONE)

Angoshtari et al. present their research focussed on understanding how the first bacterial colonisers impact the subsequent recruitment and growth of other bacteria present in river water with the aim of better understanding how pathogens might be prevented from colonising sturgeon eggs and causing disease.

This is a very well-written manuscript and I have no serious concerns with it, other than to say it might be edited for length in paces, and that the print-quality of the figures was poor.

Comments

1. P. aeruginosa PA01 is not necessarily thought of as a model for river bacteria, and it should be made clear when it is first referred to (after the Methods) that P. aeruginosa strains are regularly recovered from fresh water systems.

2. Pg. 5, line 114. The species of sturgeon should be provided.

3. Pg. 6, line 127 and elsewhere. Whenever percentages are given, w/w, w/v or v/v should be stipulated in parentheses.

4. Pg. 8, line 175. It should be stated clearly that the type of biofilm being examined and measured in this work is a submerged solid surface – liquid (A-L) interface biofilm. Other types of biofilm are equally well-studies, including air – liquid surface interface biofilms that readily form in static microtitre dish assays.

5. Pg. 9, lines 201 & 203. 5’ and 3’ should be added to the primer sequences.

6. Pg. 10, line 234. Can ‘amounts’ be contextualised – were the biofilms thicker or more dense, or did they cover more of the microtitre well walls?

7. Pg. 10, line 238. Please indicate what type of biofilm was observed in these experiments. Is it possible that some wells produced A-L interface biofilms which were not registered by the crystal violet assay because the material was washed away?

8. Table 1. Could a contingency table / Chi Square analysis be conducted to see whether species abundance is dependent on abundance in river water and in biofilms?

9. Figure 2 does not specify what type of data are shown in the graph (means ± SE?).

10. Figure 3 A & B do not show means and errors.

11. Pg. 19, line 462. This statement is disingenuous especially after hydrophobicity is noted on the following page. Do the surfaces also differ in terms of microtopography which might aid recruitment from flowing water / protection from shear force, and flexibility which may also play a role in biofilm-formation and retention?

12. Pg. 20, line 488. Can any comment be made of whether, in the microtitre wells used for the biofilm assays, there was any evidence of A-L biofilm formation, incomplete surface colonisation or possible ripping of biofilms from the surface (which may all result from aging biofilms and/or over vigorous washing and staining)?

13. Pg. 21, line 511. Some references should be provided to contextualise biofilm formation by P. aeruginosa strains isolated from fresh water and/or tested in fresh water-based medium.

6. PLOS authors have the option to publish the peer review history of their article (what does this mean?). If published, this will include your full peer review and any attached files.

Reviewer #1: No

Reviewer #2: **Yes: **Andrew J. Spiers

---

## [Author Response · Author response to Decision Letter 0]

30 Mar 2023

See Response to Reviewers attachment.

---

## [Decision Letter · Decision Letter 1]

3 May 2023

PONE-D-22-31146R1The impact of primary colonizers on the community composition of river biofilmPLOS ONE

Dear Dr. Marsh,

Thank you for submitting your manuscript to PLOS ONE. After careful consideration, we feel that it has merit but does not fully meet PLOS ONE’s publication criteria as it currently stands. Therefore, we invite you to submit a revised version of the manuscript that addresses the points raised during the review process.

We look forward to receiving your revised manuscript.

Kind regards,

Robert P Smith

Academic Editor

PLOS ONE

Journal Requirements:

Additional Editor Comments:

Thank you for submitting your revised manuscript to PLoS One. I have now heard back from both previous reviewers. From their comments, it appears that you addressed the majority of their concerns from the first round of review. However, they recommended some stylistics edits and some minor edits that should help clarify a few areas of the manuscript. Both recommended a minor revision and I agree with both of the reviewers. I look forward to receiving your revised manuscript.

Reviewers' comments:

Reviewer's Responses to Questions

**Comments to the Author**

1. If the authors have adequately addressed your comments raised in a previous round of review and you feel that this manuscript is now acceptable for publication, you may indicate that here to bypass the “Comments to the Author” section, enter your conflict of interest statement in the “Confidential to Editor” section, and submit your "Accept" recommendation.

Reviewer #1: (No Response)

Reviewer #2: (No Response)

2. Is the manuscript technically sound, and do the data support the conclusions?

Reviewer #1: Yes

Reviewer #2: Yes

3. Has the statistical analysis been performed appropriately and rigorously? 

Reviewer #1: Yes

Reviewer #2: Yes

4. Have the authors made all data underlying the findings in their manuscript fully available?

Reviewer #1: Yes

Reviewer #2: Yes

5. Is the manuscript presented in an intelligible fashion and written in standard English?

Reviewer #1: Yes

Reviewer #2: Yes

6. Review Comments to the Author

Reviewer #1: The manuscript has improved, and I had a couple of follow ups that I think need to be addressed.

Table 1:

I was referring to the lettering scheme with panels “A and B” and then the lowercase lettering scheme a, b, c, d, e being confusing and not intuitive. I still think it is. And the legend is not appropriate. There is no acknowledgement of panels A and B and a, d, and e are just results.

I don’t think it’s necessary to have it with A and B and then a. b. c. d. and e. My suggestion is to just refer to A and B only in the legend. Since everything else is labeled already describing what it is, then just include the other information in the legend without the reference to any letter. Basically, eliminate the a,b,c,d,e

“Population” is also not the current term in the table or in the legend. These are OTUs representing unique taxa… But then you use phylotypes in some of the text but populations elsewhere. Is phylotype or OTU more appropriate here? Were you screening sequences again the database to generate phylotypes or clustering sequences for OTUs? This should be clear in the methods but it wasn't to me. Table 1 says OTUs. This is an inconsistency between population, OTU, and phylotypes is throughout the manuscript.

RE: response from authors in R1:

"The cell concentration of the inoculum is irrelevant to the amount of biofilm formed in 48 hours over a wide range of initial cell concentrations. We have determined this in studies with 500-600 freshwater isolates, including those that we used in this study. "

…

"This issue is addressed above. Briefly, what is important here is the biofilm formed after 48 hours. Healthy 16-hour cultures were used as the inoculum. Each strain has its own unique

growth rate and ability to form biofilm. The biomass of biofilm produced by each strain was

strain-dependent. Under our experimental conditions the biofilm amounts as measured by crystal violet were consistent for each strain at 48 hours, over a broad range of inoculum concentration. "

Okay, but I don’t see in the text or the snippet you added these specific comments that there was consistent biofilm with a range of inoculums. Is the evidence that inoculum size of differing ranges yields the same amount of biofilm at 48 hours something you can cite from previous work or is there data? Or personal comm?

In general I find there is a lot figure and now table explanation in the text rather than just describing the text. Addressing this could make the results more concise.

Line 419 refers to supplemental material – is this suppose to be to a specific figure?

Number of references at line 579/580 seem a bit excessive.

Reviewer #2: Comments from the Reviewer of the Angoshtari et al. revised manuscript ‘The impact of primary colonizers on the community composition of river biofilm’ (PLoS ONE)

Angoshtari et al. present a revised manuscript which deals well with the comments I had made earlier. There are some very minor comments that should be addressed:

1. Line 50. It is arguable that all extant aquatic metazoans were present for millions of years; perhaps this dramatic statement can be rewritten as ‘Aquatic metazoans …’.

2. Lines 56-57. This sentence seems repetitive and could be deleted.

3. Line 101. Correct ‘I’.

4. Pseudomonas aeruginosa PA01 was isolated from an Australian patient in 1954 and it is exceedingly unlikely that the exact same strain has been isolated since then – and certainly not from the Red Cedar River. The sentence in Lines 111 – 114 needs to be altered to say that ‘other strains of this species’ are routinely isolated from fresh water sources (the rest of the manuscript needs to be checked and corrected if necessary).

5. Pseudomonas aeruginosa PA01 is referred to at various points in the manuscript but there is inconsistency in naming – a consistent approach should be adopted, e.g., Pseudomonas aeruginosa PA01 and thereafter P. aeruginosa PA01.

6. P. aeruginosa PA01 is not necessarily thought of as a model for river bacteria, and it should be made clear when it is first referred to (after the Methods) that P. aeruginosa strains are regularly recovered from freshwater systems.

7. While a particular crystal violet assay was undertaken in this work, there are various forms of the assay, including washing steps and eluent. Perhaps referring to it as ‘the standard’ (Line 203 and possibly elsewhere) is inappropriate and it should be referred to as simply ‘the crystal violet assay’.

8. Figure legends and possibly elsewhere. The term ‘average’ should be avoided as there are three possibilities; please replace with ‘mean’ throughout.

9. Line 340. The end of the sentence ‘in the biofilm that traps crystal violet’ is not needed and arguably, is incorrect: crystal violet binds to the bacterial peptidoglycan layer (and probably more generally to proteins as it stains fingers too).

10. Line 478. It is not necessary to specify ‘Earth’s’ microbiome unless there are substantive reports that microbial biomes in orbit don’t form biofilms.

11. Line 518. Delete ‘+’ as 2000 is the critical value discussed here.

7. PLOS authors have the option to publish the peer review history of their article (what does this mean?). If published, this will include your full peer review and any attached files.

Reviewer #1: No

Reviewer #2: **Yes: **Andrew J Spiers

---

## [Author Response · Author response to Decision Letter 1]

13 Jun 2023

PONE-D-22-31146R1

The impact of primary colonizers on the community composition of river biofilm

Angoshtari, Scribner & Marsh

PLOS ONE

Response to reviewers, version 2.

Below find our point-by-point response to the comments of our reviewers. 

Best regards

Terence L. Marsh, Ph.D.

Assoc. Professor Emeritus

Dept. of Microbiology & Molecular Genetics

567 Wilson Road

Michigan State University

East Lansing, MI 48824

(517)-231-6033

MARSHT@msu.edu

Reviewer #1: The manuscript has improved, and I had a couple of follow ups that I think need to be addressed.

Table 1:

I was referring to the lettering scheme with panels “A and B” and then the lowercase lettering scheme a, b, c, d, e being confusing and not intuitive. I still think it is. And the legend is not appropriate. There is no acknowledgement of panels A and B and a, d, and e are just results.

I don’t think it’s necessary to have it with A and B and then a. b. c. d. and e. My suggestion is to just refer to A and B only in the legend. Since everything else is labeled already describing what it is, then just include the other information in the legend without the reference to any letter. Basically, eliminate the a,b,c,d,e

DONE. We have eliminated the small letters and have incorporated that information into the text. 

“Population” is also not the current term in the table or in the legend. These are OTUs representing unique taxa… But then you use phylotypes in some of the text but populations elsewhere. Is phylotype or OTU more appropriate here? Were you screening sequences again the database to generate phylotypes or clustering sequences for OTUs? This should be clear in the methods but it wasn't to me. Table 1 says OTUs. This is an inconsistency between population, OTU, and phylotypes is throughout the manuscript. 

Apologies for the confusion. We now note in the M&M that our definition of OTU is set at 97% similarity to 16S rRNA. Without question this statement should have been in the first version. With the OTU set at this traditional cutoff for species-level identifications, our view is that species = population = OTU = phylotype. However, to avoid confusion, we have eliminated the usage of “phylotype”. 

Phylotype changed to…”population” in L261 L451 L457 L458 L563, and to “OTU “in L298 L300 L308 L439 L472 & L474 of Fig 6 legend L517 and eliminated from L455. 

In addition, we have reduced the usage of “species” in the manuscript. The word “species” occurs twice in the introduction where we feel it is appropriate where referring in general terms to “aquatic species”. When we are referring to a “single-species biofilm”, we feel that the term “species” is appropriate and accurate inasmuch as we are referring to a biofilm derived from a pure culture, a species of bacteria (eg. L141). We have removed the term “species” when referring to “populations/OTUs” in the bacterial community of a river. Hence “a river community comprised of hundreds of species” has been changed to “a river bacterial community comprised of hundreds of populations”. We also cite references that use the phrase “species interactions” within a biofilm. We have retained this. In line 532 we refer to the stability of our single and double species biofilms as species specific. While we think this is an appropriate term, we have changed it to “isolate specific”. In the Discussion we use “species” when referring to “all free-living species”. The term “mixed species biofilms” is common in the literature and therefore the following phrase “through a temporal sequence of species additions” seems quite appropriate. Finally, in L599 we refer to a population/OTU of Desulfobacteriaceae as a species. We have changed this to population. We feel that population is an ecologically accurate term as we use it in this manuscript, referring to a collection of individual cells belonging to the same OTU-defined group or species. We use “population” when referring to any of the previously isolated strains used in our work. For example, in line 257 we refer to “the founding populations” of biofilm. However, immediately following this remark, we have changed “between the populations forming the river biofilm” to “between the detected OTUs forming the river biofilm” for clarity.

The following was added to the M&M.

OTU identification was at the 97% similarity level of 16S rRNA using the RDP Training set. This cutoff defines what we refer to as an “OTU” or “population”, used interchangeably herein. 

Our usage of "OTU" is very limited. We used the term only 6 times in the text as indicated below (this refers to version 2 of the manuscript, the number goes up in version 3 due to replacement of all occurrences of “phylotype”). We first define OTU in the M&M. The second appearance is line 259 where we changed “population” to “OTU” in response to the reviewer. The third event is where we indicate that 10,000 OTUs were used for analysis in PAST. The remaining usage occurs in labels that are generated with MOTHUR as specific identifiers for OTUs (eg. Staphlococcus OTU00017, L310). We feel these identifiers are important to retain.

The following are where the term OTU has been retained as in the text of version 2.

L249 definition of OTU.

L253. Indicating that analysis in PAST was based on 10,000 OTUs as identified by analysis in MOTHUR.

L259. Changed “population” to “OTUs”

L310. Identification of specific OTUs identified by Mothur. 

L469. Legend Fig 6. These are specific OTU identifiers from the Mothur analysis and are quite important to the analysis.

L521. This is another specific Mothur OTU identifier. 

Again, we apologize for confusion regarding the terms OTU, population and species. We would prefer to be on the precise side of terminology but sometimes it is difficult. Two years ago, we submitted a paper on these same bacterial strains isolated from the sturgeon. We used the term “strain” throughout the paper when referencing the isolates. One reviewer argued that “strain” was inappropriate in this case because these isolates had not been “formally described”, even though we had previously presented full 16S rRNA sequence and cited information from full genome sequencing. We had to change all occurrences of “strain” to “isolate”. It’s not clear if the paper was improved.

Were you screening sequences again the database to generate phylotypes or clustering sequences for OTUs? 

We find this sentence confusing. When running this type of community analysis, one is screening sequences against a database and binning sequences into OTUs. There is no either/or here, as we view it. We would certainly agree that the definition of species is elusive, controversial and a moving target. Is it 97% rRNA similarity or is it 99% or should it be an ANI value now that we are in the genome era? We have therefore removed the term “species” wherever there might be confusion when referencing bacterial populations within our studies, as described above.

RE: response from authors in R1:

"The cell concentration of the inoculum is irrelevant to the amount of biofilm formed in 48 hours over a wide range of initial cell concentrations. We have determined this in studies with 500-600 freshwater isolates, including those that we used in this study. "

…

"This issue is addressed above. Briefly, what is important here is the biofilm formed after 48 hours. Healthy 16-hour cultures were used as the inoculum. Each strain has its own unique

growth rate and ability to form biofilm. The biomass of biofilm produced by each strain was

strain-dependent. Under our experimental conditions the biofilm amounts as measured by crystal violet were consistent for each strain at 48 hours, over a broad range of inoculum concentration. "

Okay, but I don’t see in the text or the snippet you added these specific comments that there was consistent biofilm with a range of inoculums. Is the evidence that inoculum size of differing ranges yields the same amount of biofilm at 48 hours something you can cite from previous work or is there data? Or personal comm?

Added. Our extensive experience with the bacterial strains in this investigation indicated that overnight cultures (16 hrs) used as inocula, consistently produced strain-dependent amounts of biofilm at 24 and 48 hours. This was statistically confirmed in each experiment with four replicates for each treatment

In general I find there is a lot figure and now table explanation in the text rather than just describing the text. Addressing this could make the results more concise.

This is a difficult critique to address. “…there is a lot of figure and now table explanation in the text…”. We have tried to be as concise as possible while also making sure that the reader does not get overly distant from the data. I suspect that we have a different view of what the text accompanying the figures should be. “…rather than just describing the text” We are unsure what you mean by “describing the text”. We have gone through the results section several times now to reduce what we view as unnecessary verbiage. For example we have deleted the following including the following because it was repetitious (L332 of version2) In addition, P. aeruginosa PAO1, a strong biofilm-positive control, and Serratia marcescens RL10, a soil isolate that forms a considerably less robust biofilm compared to strain PAO1, were included. Of course, eliminating the table footers necessitated adding verbiage to the results section including the following. L302 version 3 - These account for 52% of all sequences. In Table 1, note that the numbers in parentheses following the taxonomic rank indicate the confidence level of phylogenetic assignment. L307 version 3 - The top 25 OTUs identified in 24-hour biofilm account for 84% of the community. We were surprised by this objection to the table footers inasmuch as this is a common feature of tables, including those published in PLOS ONE. We hope that the current version is satisfactory.

Line 419 refers to supplemental material – is this suppose to be to a specific figure?

Specific figure number has been added. “This separation could be seen in the early samples at 4 and 8 hours as well (S4 and S5 of supplemental material).”

Number of references at line 579/580 seem a bit excessive.

We have removed 15 references and retained 13 to cover 6 genera.

Reviewer #2: Comments from the Reviewer of the Angoshtari et al. revised manuscript ‘The impact of primary colonizers on the community composition of river biofilm’ (PLoS ONE)

Angoshtari et al. present a revised manuscript which deals well with the comments I had made earlier. There are some very minor comments that should be addressed:

1. Line 50. It is arguable that all extant aquatic metazoans were present for millions of years; perhaps this dramatic statement can be rewritten as ‘Aquatic metazoans …’.

This has been changed as suggested.

2. Lines 56-57. This sentence seems repetitive and could be deleted.

Deleted

3. Line 101. Correct ‘I’.

Corrected

4. Pseudomonas aeruginosa PA01 was isolated from an Australian patient in 1954 and it is exceedingly unlikely that the exact same strain has been isolated since then – and certainly not from the Red Cedar River. The sentence in Lines 111 – 114 needs to be altered to say that ‘other strains of this species’ are routinely isolated from fresh water sources (the rest of the manuscript needs to be checked and corrected if necessary). 

Corrected. Thank you. Changed to “This strain was used as a strong positive control for biofilm formation and other strains of this species are routinely isolated from freshwater sources (44, 45, 46, 47) including the Red Cedar River (A. Chan & T.L. Marsh, unpublished), the source of river water for these studies.”

5. Pseudomonas aeruginosa PA01 is referred to at various points in the manuscript but there is inconsistency in naming – a consistent approach should be adopted, e.g., Pseudomonas aeruginosa PA01 and thereafter P. aeruginosa PA01.

DONE

6. P. aeruginosa PA01 is not necessarily thought of as a model for river bacteria, and it should be made clear when it is first referred to (after the Methods) that P. aeruginosa strains are regularly recovered from freshwater systems.

DONE. Added to L341; “While we have used P. aeruginosa PAO1 as a robust biofilm forming strain, as we point out in the Material and Methods section, strains of P. aeruginosa are routinely isolated from freshwater systems.”

7. While a particular crystal violet assay was undertaken in this work, there are various forms of the assay, including washing steps and eluent. Perhaps referring to it as ‘the standard’ (Line 203 and possibly elsewhere) is inappropriate and it should be referred to as simply ‘the crystal violet assay’.

L202, L335, L339, L558 have been changed. Line numbers are from version 2.

8. Figure legends and possibly elsewhere. The term ‘average’ should be avoided as there are three possibilities; please replace with ‘mean’ throughout.

L210, Fig 2 legend, Fig 3 legend, S2 Fig were changed.

9. Line 340. The end of the sentence ‘in the biofilm that traps crystal violet’ is not needed and arguably, is incorrect: crystal violet binds to the bacterial peptidoglycan layer (and probably more generally to proteins as it stains fingers too).

Deleted.

10. Line 478. It is not necessary to specify ‘Earth’s’ microbiome unless there are substantive reports that microbial biomes in orbit don’t form biofilms.

CHANGED TO “For example, we do not know if all free-living species of bacteria can form biofilms, but it has become apparent that a substantial fraction of bacteria in the environment exists as biofilm (67).” We have changed this but still feel that using the phrase “earth’s microbiome” does indicate the totality of bacteria on the planet and does not necessarily imply bugs anywhere else in the cosmos.

11. Line 518. Delete ‘+’ as 2000 is the critical value discussed here.

Done

---

## [Editor Report · Decision Letter 2]

19 Jun 2023

The impact of primary colonizers on the community composition of river biofilm

PONE-D-22-31146R2

Dear Dr. Marsh,

We’re pleased to inform you that your manuscript has been judged scientifically suitable for publication and will be formally accepted for publication once it meets all outstanding technical requirements.

Kind regards,

Robert P Smith

Academic Editor

PLOS ONE

Additional Editor Comments: Thank you for addressing all of the minor comments raised by both reviewers.

---

## [Editor Report · Acceptance letter]

26 Jun 2023

PONE-D-22-31146R2 

The impact of primary colonizers on the community composition of river biofilm 

Dear Dr. Marsh:

I'm pleased to inform you that your manuscript has been deemed suitable for publication in PLOS ONE. Congratulations! Your manuscript is now with our production department. 

Kind regards, 

on behalf of

Dr. Robert P Smith 

Academic Editor

PLOS ONE